# In Vitro Investigation of Vascular Permeability in Endothelial Cells from Human Artery, Vein and Lung Microvessels at Steady-State and Anaphylactic Conditions

**DOI:** 10.3390/biomedicines9040439

**Published:** 2021-04-19

**Authors:** Katrine T. Callesen, Alma Yuste-Montalvo, Lars K. Poulsen, Bettina M. Jensen, Vanesa Esteban

**Affiliations:** 1Laboratory of Medical Allergology, Copenhagen University Hospital at Gentofte, DK-2900 Hellerup, Denmark; katrinecallesen@hotmail.com (K.T.C.); lkpallgy@mail.dk (L.K.P.); bettina.margrethe.jensen@regionh.dk (B.M.J.); 2Department of Allergy and Immunology, IIS-Fundación Jiménez Díaz, UAM, 28040 Madrid, Spain; almamonyu@gmail.com; 3Faculty of Medicine and Biomedicine, Alfonso X El Sabio University, 28691 Madrid, Spain; 4Red de Asma, Reacciones Adversas y Alérgicas (ARADyAL), Instituto de Salud Carlos III, 28029 Madrid, Spain

**Keywords:** endothelial cells, vascular permeability, histamine, PAF, thrombin, macromolecular tracer assay

## Abstract

Human anaphylactic reactions largely involve an increase in vascular permeability, which is mainly controlled by endothelial cells (ECs). Due to the acute and serious nature of human anaphylaxis, in vivo studies of blood vessels must be replaced or supplemented with in vitro models. Therefore, we used a macromolecular tracer assay (MMTA) to investigate the EC permeability of three phenotypes of human ECs: artery (HAECs), vein (HSVECs) and microvessels from lung (HMLECs). ECs were stimulated with two fast-acting anaphylactic mediators (histamine and platelet-activating factor (PAF)) and one longer-lasting mediator (thrombin). At steady-state conditions, HSVEC monolayers were the most permeable and HMLEC the least (15.8% and 8.3% after 60 min, respectively). No response was found in ECs from artery or vein to any stimuli. ECs from microvessels reacted to stimulation with thrombin and also demonstrated a tendency of increased permeability for PAF. There was no reaction for histamine. This was not caused by missing receptor expression, as all three EC phenotypes expressed receptors for both PAF and histamine. The scarce response to fast-acting mediators illustrates that the MMTA is not suitable for investigating EC permeability to anaphylactic mediators.

## 1. Introduction

The permeability of blood vessels is mainly controlled by endothelial cells (ECs) that line the inside of all blood vessel walls as a monolayer of cells that tightly control what solutes and cells are allowed to pass between the blood stream and surrounding tissue. ECs are highly heterogeneous and display different susceptibilities to a number of stimulants, including inflammatory and vasoactive agents [1,2]. As vascular permeability is involved in many physiological and pathophysiological inflammatory processes, EC monolayer stability has been extensively studied both in vivo, in animal models, and in vitro using a number of different methods [3,4].

A classical method to investigate EC permeability is through the use of a macromolecular tracer assay (MMTA). Here, cells are seeded in a Transwell^®^ insert, containing a highly porous membrane upon which the ECs will form a monolayer, thereby mimicking their structure in vivo. EC permeability can then be assessed by the passage of, for example, fluorophore-coupled beads, from the inner to the outer chamber [5,6,7].

Anaphylaxis (AX) is the most severe manifestation of allergy, where life-threatening symptoms rapidly develop. The current understanding of the AX pathophysiology greatly involves the cardiovascular system [8], with reports on losses of up to 35% of the blood volume into the extravascular space within 10 min of allergen exposure [9]. Fast-acting mediators such as histamine and platelet-activating factor (PAF) are likely to be some of the main mediators involved in this rapid increase in vascular permeability associated with AX [4,10,11,12,13]. Mouse studies have shown that there is a heterogeneous response in the permeability of different types of blood vessels in response to stimuli [4]. However, it is not known whether this heterogeneity is found among human EC phenotypes. To investigate this, in vitro assays are needed. Since AX often involves fast-acting mediators, it is important to evaluate the usefulness of the specific in vitro assay. Therefore, the aim of this study was to use an MMTA to investigate EC permeability at steady-state and after contact with the fast-acting mediators histamine and PAF, using three different human EC phenotypes: human arterial ECs (HAECs), human saphenous vein ECs (HSVECs) and human microvascular lung ECs (HMLECs).

## 2. Materials and Methods 

### 2.1. Endothelial Cell Origin

HMLECs were commercially procured and maintained in EBM-2 Complete Medium (EBM-2 medium + EGM-2 supplement + 100 µg/mL heparin) + 30 µg/mL Endothelial Cell Growth Factor (ECGF) + 20% Fetal Bovine Serum (FBS).

HSVECs and HAECs were isolated as previously described [13]. Briefly, the endothelial side of vein pieces surgically removed due to varicosity or entire arterial segments from aortae of patients undergoing cardiac surgery were incubated in DMEM/F-12 medium with 0.1% Collagenase I (Gibco, Thermo Fisher Scientific, UK) for 30 min (for HSVECs) or overnight (for HAECs). After incubation, the reaction was stopped and the ECs were scraped off the vessel walls, centrifuged at 500× *g* for 5 min at RT and resuspended in DMEM/F-12 Complete Medium (DMEM/F-12 + 100 U/mL penicillin + 100 µg/mL streptavidin + 2 mM L-glutamine + 2.5 µg/mL Amphotericin B + 100 µg/mL heparin + 30 µg/mL ECGF + 20% FBS). After 2–5 days of expansion, the HSVECs and HAECs were detached from the culture plates using 0.025% trypsin-EDTA and isolated through positive selection using 2.5 µg/mL mouse anti-human CD31 antibody (BD Biosciences-EU) followed by 2 × 10^6^ beads/mL magnetic beads coated with anti-mouse IgG (Invitrogen, Thermo Fisher Scientific, UK). Both incubation steps were made for 30 min at 4 °C. The cell suspensions were then passed through a column placed in a magnet, ensuring the isolation of HSVECs or HAECs. The isolated ECs were resuspended in Complete Medium and cultured on 0.5% gelatin-coated plates. The study was approved by the Research Ethics Committee at Hospital Universitario Fundación Jiménez Díaz (approval number PIC076-18_FJD). Informed consent was obtained from all patients.

HMLECs and HAECs were immortalized with lentivirus (Lenti-SV40-Ta, Capital Biosciences Inc., Gaithersburg, MD, USA) according to the manufacturer’s instructions. All cell cultures were expanded and maintained at 37 °C, 5% CO_2_.

### 2.2. Macromolecular Tracer Assay

Transwell^®^ inserts (6.5 mm insert, 0.4 µm, Corning Life Science, USA) were incubated in 24-well plates with FBS for 30 min at RT. Next, a coating of the membrane was applied with 0.5% of gelatin for HSVECs and HAECs. After this, they were moved to a 24-well plate with 600 µL Complete Medium per well, and 10^5^ cells (HSVECs, HAECs, or HMLECs) in 200 µL Complete Medium was added to the inner chamber. Plates were incubated overnight before transwells were moved to Starvation Medium (for HAECs and HSVECs: DMEM/F-12 + 100 U/mL penicillin + 100 µg/mL streptavidin + 2 mM L-glutamine + 2.5 µg/mL Amphotericin B + 0.5% FBS; for HMLECs: EBM-2 + EGM-2 supplement + 0.5% FBS) and incubated overnight.

The MMTA was performed by moving transwells to a new plate with Pure Medium (for HAECs and HSVECs: DMEM/F-12 + 100 U/mL penicillin + 100 µg/mL streptavidin + 2 mM L-glutamine + 2.5 µg/mL Amphotericin B; for HMLECs: EBM-2 + EGM-2 supplement) and adding 100 µL FITC-Dextran solution (1 mg/mL of 40 kDa FITC-Dextran) in Pure Medium, either with 10 µM PAF, 10 µM histamine, 0.1 U/mL thrombin, or unstimulated (“basal sample”) to the inner chamber of all transwells. The plate was incubated for 5 min (timer started at the first well) before moving transwells to the next plate for 25 min, and then transwells were moved to a final plate for 30 min. All sample tests were conducted at least in duplicate. After incubation, the presence of FITC in the outer chamber (i.e., the well of the plate) was measured in a black 96-well plate on a fluorescence plate reader (Infinite 200 PRO, Tecan; ex: 485 nm, em: 535 nm). Data are depicted as fold change from basal sample (for stimulation experiments) or as the median of measured fluorescence values of extravasated FITC (for basal samples). Figure 1 schematically illustrates the process. An identical experiment was run to test for passive diffusion, with the same setup as described for basal samples but without any ECs added (“Blank”).

### 2.3. SDS-PAGE

Protein was extracted from confluent EC cultures by adding lysis buffer (Tris-HCl (50 mM), NaCl (0.15 M), EDTA (2.5 mM), EGTA (2 mM), 0.2% Triton X-100, 0.3% NP40) + 1% DTT + 0.5% PMSF + 0.05% protease inhibitors, scraped off the Petri dish and transferred to a tube. The tube was centrifuged at 5000 × *g* for 15 min at 4 °C and the pellet was discarded. Samples were boiled for 5 min with 4× LDS NuPAGE sample buffer, centrifuged, and loaded onto a NuPAGE 4–12% Bis-Tris gel. The gel was run for 1 h using MOPS buffer. 

### 2.4. Western Blot

The gel was blotted onto a nitrocellulose membrane, washed with TBST buffer followed by 1 h blocking (RT) of unspecific binding using either TBST + 4% Bovine serum albumin (BSA) for H1R and H2R, or TBST + 3% skimmed milk powder for PTAFR. Membranes were then incubated with primary antibody (Mouse anti-H1R (1:100), Rabbit anti-H2R (1:500) (GeneTex, Alton Pkwy Irvine, USA), Rabbit anti-PTAFR (1:200) (Cayman Chemical, Michigan, USA)) overnight at 4 °C, washed and incubated with a secondary antibody (Goat anti-rabbit IgG-HRP (1:3000) or Goat anti-mouse IgG-peroxidase (1:10,000)) for 1 h at RT. Antibody dilutions were made in TBS + 4% BSA. After this, membranes were washed, and proteins were examined on X-ray film using ECL HRP chemiluminescence. Membranes were re-probed using mouse anti-β-actin (1:80,000) and goat anti-mouse IgG-peroxidase (1:10,000). 

Protein band intensity was determined using ImageJ 1.52v and relative protein expression was calculated by normalizing protein expression to β-actin.

### 2.5. Reagents

Products were acquired in the following companies: HMLECs, EBM-2 medium and supplements, DMEM/F-12 medium, Amphotericin B and trypsin-EDTA from Lonza Walkersville, Inc. USA. Heparin, ECGF, FBS, penicillin, streptavidin L-glutamine, thrombin, FITC-Dextran, PAF, histamine, thrombin BSA, goat anti-mouse IgG-peroxidase and anti-β-actin from Merck Life Science S.L.U, Spain. Anti-H1R and goat anti-rabbit IgG-HRP from Santa Cruz Biotechnology, Inc., Texas, USA. NuPAGE reagents, nitrocellulose membrane and ECL HRP chemiluminescence from Invitrogen, Thermo Fisher Scientific, Slangerup, DK).

### 2.6. Statistics

Where appropriate, Mann–Whitney test was used. All graphs are illustrated in GraphPad Prism 8.3.0 with data expressed as median values with interquartile range. *p*-values < 0.05 are shown.

## 3. Results

ECs are known to express receptors for both PAF and histamine in vivo, but the high plasticity of ECs in vitro led us to verify the expression of the histamine receptors H1R and H2R, and the PAF receptor PTAFR in the three cultured EC phenotypes (HAECs, HSVECs, and HMLECs). All three EC phenotypes expressed H1R (Figure 2A), H2R (Figure 2B), and PTAFR (Figure 2C) at a comparable level (Figure 2D–F).

For the MMTA to function, ECs must form a functional monolayer in the inner chamber of the transwells. The strength of the endothelial barrier can be measured by their ability to block the passage of larger molecules to the outer chamber. This ability was apparent when comparing extravasation of FITC-Dextran beads in blank transwells (i.e., without any cells to uphold a barrier, thereby allowing for passive diffusion) to transwells containing monolayers of basal (unstimulated) HSVECs, HAECs, or HMLECs (Figure 3A). in vivo, veins are leakier than arteries and microvessels. When comparing the basal permeability of the EC monolayers from different EC phenotypes in the MMTA in vitro system, we observed a similar pattern, where HMLECs and HAECs formed the tightest monolayers (8.3% and 10.2% of blank after 60 min, respectively) while HSVECs produced a leakier monolayer (15.8% of blank after 60 min) (Figure 3B). This indicates that ECs, despite being in an in vitro environment, might still retain some of their original phenotypic behavior.

Next, we investigated the monolayer permeability of all three EC phenotypes to stimulation with the two fast-acting mediators histamine and PAF, as well as the longer-lasting mediator thrombin. Interestingly, all the mediators used failed to induce a detectable increase in permeability in HSVECs and HAECs (Figure 4A,B). For HAECs, histamine even appeared to mediate an opposite effect. The HMLECs, however, showed a clear increase in permeability to thrombin, moderate to PAF, and just a tendency with histamine (Figure 4C).

## 4. Discussion

Molecular mechanisms underlying AX are very difficult to investigate in humans as the reaction often takes place outside of clinical settings and can be very rapid in onset. Due to the acuteness and dangerous nature of AX, most studies addressing the mechanisms behind it cannot be conducted in humans and are instead conducted through in vitro methods or through animal models [10,12]. Vascular permeability is one of the pathophysiological features commonly present in AX and can lead to some of the more life-threatening consequences, such as hypoxia and severe drop in blood pressure [10,13]. To better understand the mechanisms behind this, it is therefore of interest to apply in vitro methods that allow us to monitor relevant responses of human ECs in relation to AX stimuli. Two of the mediators known to influence vascular permeability during AX are histamine and PAF, and we therefore focused on these two in the present study. As a control we also included thrombin, as this is considered to be an overall inducer of permeability in vitro [2].

In AX, the whole circulatory system is disturbed but the main type of blood vessels involved in the severity of the reactions is not completely understood [8]. in vivo, vascular permeability is mainly thought to arise in the postcapillary venules, although capillaries and large veins may also be affected in severe situations [1,2,3]. Egawa et al. used a mouse model of anaphylaxis to evaluate the direct effect of histamine injection, or in a passive systemic anaphylaxis, on the microvascular system. Here they showed that the increased permeability mainly arose from the venules [4]. However, the effects on arteries and veins were not examined.

It is known that ECs have a high plasticity, and differences in in vitro behavior, compared to in vivo, have been observed [1,2,14]. Despite this, we demonstrated that the EC phenotypes used in our study all retained some of the expected behaviors; they expressed specific receptors to both PAF and histamine that have also been found on ECs in vivo [15,16] (Figure 2), and they were able to form monolayers that separated the inner chamber from the outer chamber (Figure 3A). In this in vitro system, we also observed that HSVECs from veins presented a higher leakiness than both HAECs from arteries and HMLECs from microvessels (Figure 3B). Accordingly, in vivo studies of mouse arteries and veins show a similar pattern as observed here, where arteries have much lower permeability than veins [17]. On the other hand, as the microvasculature in vivo tightly controls the permeability processes during inflammatory situations, these are expected to have low permeability at steady-state [18]. Both of these observations correspond to the results observed in our in vitro settings.

In this study, we investigated the permeability of human EC monolayers using an MMTA. We failed to detect any response to histamine in the three EC phenotypes, while increased permeability to PAF and thrombin was only observed in HMLECs. We have previously shown that 1 μM histamine or 0.1 μM PAF can mediate the permeability of human dermal microvascular ECs [13]; thus, we believe that the concentrations used in this study are not too low. Due to the overall low responses observed, we evaluated the passive diffusion speed of the FITC-Dextran beads in the MMTA without any ECs seeded. Here we observed a very slow diffusion rate, with only 3.06% ± 0.82% within 5 min and 54.45% ± 11.35% after 60 min (data not shown), indicating that this assay is not ideal to test fast and transient effects.

Such fast and transient increases in permeability have been observed in in vitro studies on histamine, where its effects were visible within the first 30 s after stimulation and finished, or perhaps even the opposite, after 5 min [6,19]. In a previous study, we observed this quick response in human dermal microvascular ECs, where increased permeability was observed to histamine within five minutes, but no effect at later timepoints. In contrast, HSVECs and HAECs, had no effects (and even the opposite for HAECs) resulting from the addition of histamine [13]. This fact, combined with the slow diffusion rate observed in the MMTA, could explain why none of the EC phenotypes demonstrated detectable increased permeability in this system (Figure 4A–C) even though they all expressed both the H1R and H2R receptors.

In vitro assays have shown that PAF elicits a longer period of activation than that observed for histamine [7,15]. This could explain why HMLECs had a detectable increase in permeability to PAF, even though none was observed to histamine (Figure 4C). These results correspond to our previous findings obtained using ECs from dermal microvasculature [13], and could indicate a general tendency of PAF activation in ECs from microvessels. However, no effect was observed in either HSVEC or HAEC cultures in response to PAF. This lack of increased EC permeability both in HAECs and HSVECs, but not in HMLECs, indicates that the increased permeability occurring in blood vessels during human AX might occur mainly in the microvessels.

Surprisingly, HSVEC and HAEC cultures also failed to respond to thrombin (Figure 4A,B). Therefore, another possible explanation for the lack of response to all stimuli tested in these two EC phenotypes could be that primary ECs derived from human arteries and veins may not be suitable for permeability studies in general. in vivo, blood vessels are not exclusively an EC monolayer, and the surrounding layers (among others, containing smooth muscle cells and pericytes) influence the permeability process through the interplay with the ECs [20]. We therefore also attempted to include smooth muscle cells in the setup of transwells for both HAEC and HSVEC cultures, but this had no effect on the observed results (data not shown). Besides supporting cells, the MMTA also lacked other factors known to affect the behavior of vascular permeability in vivo, including the glycocalyx, basement membrane, and shear stress from blood flow. All of these factors could very well influence the lack of change in permeability observed in both HSVEC and HAEC experiments. However, the overall low diffusion rate observed in the MMTA will still most likely make this method suboptimal in situations with fast-acting mediators, as is found in AX. Possible alternative methods for investigating EC permeability to AX mediators should have fast and sensitive detection to measure the transient changes that might occur. One such is the transendothelial electrical resistance (TER), although this has the best effect on monolayers with very low permeability (as is seen for epithelial cells or ECs from the blood–brain barrier) or the EndOhm system, which has lower variation than TER. Finally, systems where single-cell morphological changes can be measured, such as xCELLigence or ECIS [6], are both sensitive and rapid in detection and may prove useful to detect differences between EC phenotypes to AX mediators.

In conclusion, our study shows heterogeneity between different phenotypes of human ECs in vitro. At steady-state, ECs from veins are more permeable than ECs from both arteries and microvessels—an observation that corresponds to the expected in vivo behavior. In response to stimuli, we found that microvascular ECs had a higher response to mediators than ECs from both arteries and veins. In addition, it appears that the MMTA is not an optimal system to study fast-acting changes associated with endothelial permeability, and alternative methods are needed to further investigate the molecular mechanisms behind vascular permeability during AX.

## Figures and Tables

**Figure 1 biomedicines-09-00439-f001:**
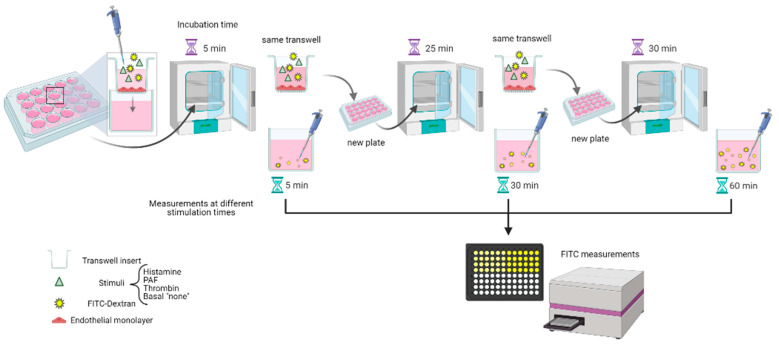
Diagram showing the flow of work for the macromolecular tracer assay. Created with BioRender.com.

**Figure 2 biomedicines-09-00439-f002:**
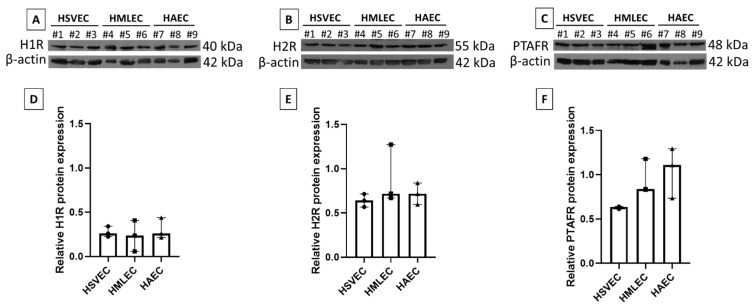
Endothelial receptor expression. Human saphenous vein endothelial cells (HSVEC#1–3), human microvascular lung endothelial cells (HMLEC#4–6), and human arterial endothelial cells (HAEC#7–9) were investigated for histamine 1 receptor, H1R (**A**,**D**), histamine 2 receptor, H2R (**B**,**E**), and platelet activating factor receptor, PTAFR (**C**,**F**). (**A**–**C**) Western blots of specific protein and loading control (β-actin). (**D**–**F**) Relative protein expression (normalized to β-actin). All were tested in three independent cell cultures (#1–3).

**Figure 3 biomedicines-09-00439-f003:**
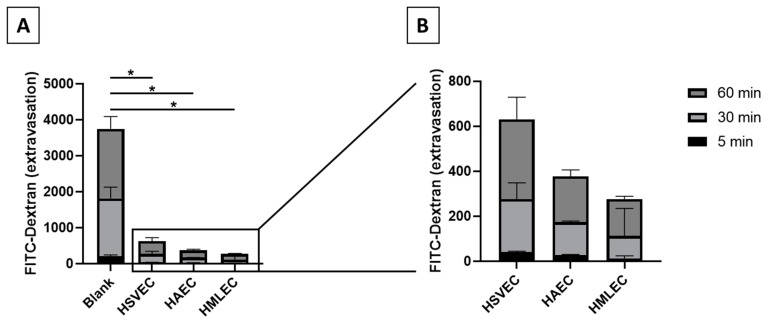
Extravasation of FITC-Dextran in macromolecular tracer assay. (**A**) The extravasation of FITC-Dextran was measured in transwells without cells (“Blank”) and compared to transwells with monolayers of basal (unstimulated) ECs. (**B**) Magnification of basal diffusion experiments. Blank, *n* = 10; Human saphenous vein endothelial cells (HSVEC), *n* = 5; human arterial endothelial cells (HAEC), *n* = 4; and human microvascular lung endothelial cells (HMLEC), *n* = 5. * *p* < 0.01.

**Figure 4 biomedicines-09-00439-f004:**
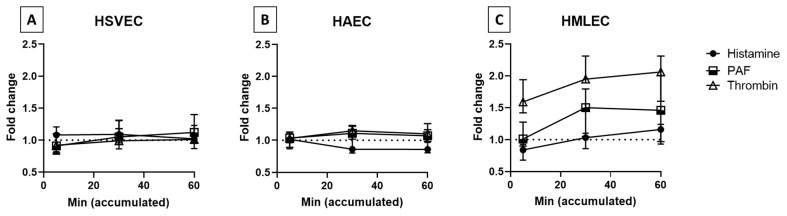
EC permeability upon stimulation with histamine, platelet activator factor (PAF), and thrombin. (**A**) Human saphenous vein endothelial cells (HSVEC; *n* = 5 for histamine and PAF, *n* = 3 for thrombin), (**B**) human arterial endothelial cells (HAEC; *n* = 4 for histamine and PAF, *n* = 3 for thrombin), and (**C**) human microvascular lung endothelial cells (HMLEC; *n* = 5 for histamine and PAF, *n* = 3 for thrombin) were either stimulated with 10 µM histamine, 10 µM PAF, or 0.1 U/mL thrombin, or were unstimulated (basal). Level of FITC was measured from the outer chamber at the indicated accumulated time. Data are depicted as fold change from the basal samples. Dotted line: basal sample indication.

## Data Availability

Data can be available upon request.

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
