# Peer review of "In Vitro Investigation of Vascular Permeability in Endothelial Cells from Human Artery, Vein and Lung Microvessels at Steady-State and Anaphylactic Conditions"

_biomedicines, 2021, doi:10.3390/biomedicines9040439_

Round 1
Reviewer 1 Report
The authors tested endothelial cell permeability in vitro using 3 different types of cells using MMTA assay. They reported that the expression of receptors to the stimulators was comparable in the 3 types of cells, and cells retained some of their original phenotype behavior in terms of permeability in vitro. With MMTA, HSVEC and HAEC did not respond to the stimulation with histamine, PAF, and thrombin while the permeability of HMLEC increased upon the stimulation with histamine, PAF, and thrombin, respectively.
Comments:
1. In the explanation of the failed responses in HSVEC and HAEC cells, the authors discussed that the MMTA was not an ideal assay to test fast and transit effects. However, Figure 1A showed that passive diffusion was several folds higher in blank than the diffusion in assays with cells within 60 min assay, suggesting that the passive diffusion rate was high enough. The authors used 10uM histamine, 10 uM PAF, and 0.1 u/ml thrombin in the Fig3 experiment. I wonder if the concentrations of histamine, PAF, and thrombin were too low to stimulate responses in vitro for those cells? It would be nice to establish dose-response curves accompanied by viability assays to optimize the treatment concentration for histamine, PAF, and thrombin.
2. Were there changes in protein levels of H1R, H2R and PTAFR in HSVEC, HAEC and MMTA cells in response to the treatments of histamine, PAF and thrombin?
3. In Fig 2, is “FITC-Dextran” on the Y axel fluorescence values?
Author Response
The authors tested endothelial cell permeability in vitro using 3 different types of cells using MMTA assay. They reported that the expression of receptors to the stimulators was comparable in the 3 types of cells, and cells retained some of their original phenotype behavior in terms of permeability in vitro. With MMTA, HSVEC and HAEC did not respond to the stimulation with histamine, PAF, and thrombin while the permeability of HMLEC increased upon the stimulation with histamine, PAF, and thrombin, respectively.
Comments:
Point 1: In the explanation of the failed responses in HSVEC and HAEC cells, the authors discussed that the MMTA was not an ideal assay to test fast and transit effects. However, Figure 1A showed that passive diffusion was several folds higher in blank than the diffusion in assays with cells within 60 min assay, suggesting that the passive diffusion rate was high enough. The authors used 10uM histamine, 10 uM PAF, and 0.1 u/ml thrombin in the Fig3 experiment. I wonder if the concentrations of histamine, PAF, and thrombin were too low to stimulate responses in vitro for those cells? It would be nice to establish dose-response curves accompanied by viability assays to optimize the treatment concentration for histamine, PAF, and thrombin.
Response 1: We acknowledge the comment regarding the passive diffusion rate. It is true that diffusion takes place in an empty well (“Blank value in previous Figure 2”), however, it happens at a very slow diffusion rate i.e. when compared to the total amount of FITC-dextran load in the upper chamber this only accounts for 3.06%± 0.82% within 5 minutes and 26.5%±5,6% within 30 min (n=10). When applying a monolayer of cells this will naturally limit the passive diffusion even when permeability is induced. Furthermore, from previous figure 3, it seems that the “frame/window” for leakiness when stimulating HMLECs is between 5-30 min (this is the peak where FITC values in fold changes is seen). Thus, prolonging the diffusion time does not benefit the result. Therefore, we believe that the dynamic of the assay is not good enough to detect changes in permeability which appears rather fast and possible transient.
We agree with the reviewer that optimal concentrations of the stimuli need to be confirmed. We tested the following concentrations on all three EC phenotypes: 0.1 µM, 1 µM and 10 µM for histamine, 1 µM, 10 µM and 100 µM for PAF and 0.1 U/ml and 0.5 U/ml for Thrombin. In this, we found the optimal concentrations for HMLECs to be 10 µM for histamine and PAF and 0.1 U/ml for thrombin. No reaction was seen with HAEC and HSVEC. We have previously shown that 1 µM histamine or 0.1 µM PAF can mediate permeability of human dermal microvascular ECs (see Ballesteros et al., 2017) thus we believe that the concentrations used in this study is not too low. This comment has now been added to the discussion.
Point 2: Were there changes in protein levels of H1R, H2R and PTAFR in HSVEC, HAEC and MMTA cells in response to the treatments of histamine, PAF and thrombin?
Response 2: We appreciate your comment but we only looked at receptor expression in resting cells to confirm their potential reacting to the applied stimuli. The expression might very well change after stimulation as receptor expression often is regulated by its ligand, however, it was not the scope of this study to investigate the receptor dynamic in ECs but to address their vascular permeability.
Point 3: is “FITC-Dextran” on the Y axel fluorescence values?
Response 3: The FITC-dextran are fluorescence values. It is also indicated in the method section “Data is depicted as fold change from basal sample (for stimulation experiments) or as the median of measured fluorescence values of extravasated FITC (for basal samples).”

Reviewer 2 Report
The article by Callesen et al entitled “In vitro investigation of vascular permeability in endothelial cells from human artery, vein and lung micro-vessels at steady-state and anaphylactic conditions”. The authors aim to evaluate the efficiency of a classical method with a macromolecular tracer assay (MMTA) used to investigate endothelial cell (EC) permeability, in assessing the permeability of different type of human EC at steady state and after stimulation with anaphylaxis mediators. They used microvascular and macrovascular ECs. This method was previously used by the same team to compare the same macrovascular ECs to human dermal microvascular ECs. I do not understand why results are different for macrovascular ECs between both study. Overall, the article is not as clear as the previous one from the same team, and would clearly benefit from re-writing in order to highlight the results.
Specific comments:
- I do not understand: line 25: “Overall, data points to a higher permeability of human microvessels to anaphylactic mediators”, whereas line 167: “histamine caused no detectable effect in any EC … Thrombin showed clear increase in permeability for HMLEC”. PAF only caused a tendency of increased permeability, and thrombin is not an AX mediator. All this information contradicts…
- Please provide schema for the MMTA (methods 2.2): transwell are moving to so many different plates that it is difficult to follow how the experiment is performed
- Line 100: “All samples were at least texted? in duplicate.” Please correct to “tested”
- Line 108, please correct to ?????r?? ???? ????? ?? ????? ?â„Ž?????
- Line 148: “For the MMTA to function ECs must form a functional monolayer in the inner chamber of the transwells”, may be a coma is needed between functions and ECs?
- Have the authors performed a dose response to find the best dose of histamine and PAF to be used in in vitro model?
- How do the authors explain the difference between results obtained with dermal and lung microvascular ECs (comparison with ref 13 from the same authors)?
Author Response
The article by Callesen et al entitled “In vitro investigation of vascular permeability in endothelial cells from human artery, vein and lung micro-vessels at steady-state and anaphylactic conditions”. The authors aim to evaluate the efficiency of a classical method with a macromolecular tracer assay (MMTA) used to investigate endothelial cell (EC) permeability, in assessing the permeability of different type of human EC at steady state and after stimulation with anaphylaxis mediators. They used microvascular and macrovascular ECs. This method was previously used by the same team to compare the same macrovascular ECs to human dermal microvascular ECs. I do not understand why results are different for macrovascular ECs between both study. Overall, the article is not as clear as the previous one from the same team, and would clearly benefit from re-writing in order to highlight the results.
Author response:
We appreciate the reviewer’s comments and we have modified the manuscript by adding a new figure 1 illustrating the MMTA method and re-written paragraphs. We hope that this together with our reply to the reviewer´s concern will illustrate more clearly the findings.
Next, we will specifically clarify the reviewer´s concern on” why results are different for macrovascular ECs between both study “in the following section.
We have never observed an increase in vascular permeability for macrovascular ECs in any of our studies. Our previous results did not show modification in barrier properties after stimulation with histamine in HVECs or HAECs at short periods of time (5min). This is a result in line with those that we show here. However, in the previous study we observed a cell-dilating effect (i.e., opposite reaction to permeability) induced by histamine at a longer incubation time point (30minutes), (see figure 4C, Ballesteros et al., 2017). Moreover, a tendency toward this same dynamic was seen in HAECs. This cell dilating effect (vs the basal stage of each cell type) is not statically supported in the current article, and we agree with the reviewer on this aspect, but the trend is seen in HAEC with Histamine after 30 min stimulation.
Altogether, we believe that our data in this study and the previous one points in the same direction i.e., microvascular ECs (HMLEC and HMDEC) are susceptible to increase vascular permeability responding to mediators at short times, whereas macrovascular ECs (HSVEC and HAEC) rather illustrate a counter action due to a cell-dilation.
Specific comments:
Point 1: I do not understand: line 25: “Overall, data points to a higher permeability of human microvessels to anaphylactic mediators”, whereas line 167: “histamine caused no detectable effect in any EC … Thrombin showed clear increase in permeability for HMLEC”. PAF only caused a tendency of increased permeability, and thrombin is not an AX mediator. All this information contradicts…
Response 1: We appreciate the reviewer's comment addressing this aspect, and in order to clarify it, the main text has been modified accordingly to our conclusion. The following is now stated in the result section: Interestingly, all the mediators used failed to induce a detectable increase in permeability in HSVEC and HAEC (Figure 4A-B). For HAEC, histamine even appeared to mediate an opposite effect. The HMLEC, however, showed a clear increase in permeability to Thrombin, moderate to PAF and a tendency with Histamine (Figure 4C). In addition, we have now deleted the sentence in the abstract (line 25).
Point 2: Please provide schema for the MMTA (methods 2.2): transwell are moving to so many different plates that it is difficult to follow how the experiment is performed
Response 2: In accordance with your suggestion, we have now outlined the steps in the MMTA by adding a new figure 1, in the methodology section, illustrating the assay.
Point 3: Line 100: “All samples were at least texted? in duplicate.” Please correct to “tested”
Response 3: We apologize for the oversight. This is now corrected.
Point 4: Line 108, please correct to ?????r?? ???? ????? ?? ????? ?â„Ž?????
Response 4: We apologize for the oversight. This is now corrected.
Point 5: Line 148: “For the MMTA to function ECs must form a functional monolayer in the inner chamber of the transwells”, may be a coma is needed between functions and ECs?
Response 5: Thank you for the suggestion, we agree and have corrected accordingly
Point 6: Have the authors performed a dose response to find the best dose of histamine and PAF to be used in in vitro model?
Response 6: We agree with the reviewer's appraisal and confirm that we tested the following concentrations on all three EC phenotypes: 0.1 µM, 1 µM and 10 µM for histamine, 1 µM, 10 µM and 100 µM for PAF and 0.1 U/ml and 0.5 U/ml for Thrombin. In this, we found the optimal concentrations for HMLECs to be 10 µM for histamine and PAF and 0.1 U/ml for thrombin. No reaction was seen with HAEC and HSVEC. We have previously shown that 1 µM histamine or 0.1 µM PAF can mediate permeability of human dermal microvascular ECs (see Ballesteros et al., 2017) thus we believe that the concentrations used in this study is not too low. This comment has now been added to the discussion.
Point 7: How do the authors explain the difference between results obtained with dermal and lung microvascular ECs (comparison with ref 13 from the same authors)?
Response 7: We again appreciate the reviewer's constructive feedback, as it refers to important aspects of our aims in order to gain knowledge in anaphylaxis. ECs possess a huge heterogeneity as described in (1,2). EC heterogeneity in our field of interest also means that microvessels in lung and dermis may not necessarily respond identically to anaphylactic mediators. In Ballesteros et al., 2017, reference 13), it is shown the increase in vascular permeability in dermal microvascular ECs to both histamine and PAF after 5 minutes while the response to histamine disappears again before next timepoint (30 minutes) where PAF response persists, although not significant. In the current study we do not find any significant change to either PAF or histamine after 5 minutes, but at later timepoints we find a similar tendency of increase to PAF as observed for dermal microvessels. These differences may indicate that the dermal microvascular ECs respond faster than lung microvascular ECs. Again, further studies designed with the purpose to compare both types of microvasculature in response to mediators will improve the knowledge about a different response of the endothelial cells and the vascular niche in anaphylaxis.
REFERENCES:
- Aird WC. Endothelium and haemostasis. Hamostaseologie. 2015;35(1):11-6. doi: 10.5482/HAMO-14-11-0075.
- Aird WC. Endothelial cell heterogeneity. Crit Care Med. 2003 Apr;31(4 Suppl):S221-30.

Round 2
Reviewer 1 Report
The authors responded to my comments properly and no more comments.